# Effect of TiO$_2$ Addition on the Melting Behaviors of CaO-SiO$_2$-30%Al$_2$O$_3$-5%MgO System Refining Slags

Xiaomeng Zhang [1,2], Ziwen Yan [1,2], Zhiyin Deng [1,2,*] and Miaoyong Zhu [1,2]

1   Key Laboratory for Ecological Metallurgy of Multimetallic Mineral (Ministry of Education), Northeastern University, Shenyang 110819, China
2   School of Metallurgy, Northeastern University, Shenyang 110819, China
*   Correspondence: dengzy@smm.neu.edu.cn

**Abstract:** To improve the yield of titanium alloy, a certain amount of TiO$_2$ can be added to the refining slag system of Ti-bearing steel grades. With the aim of understanding the effect of TiO$_2$ addition on the melting behaviors of CaO-SiO$_2$-30%Al$_2$O$_3$-5%MgO refining slags, the melting points of the slags and the phases in the slags are herein studied at different temperatures in the laboratory. It is found that with the increase in TiO$_2$ content (0~10%) in slag, the melting point of the slags drops first, and then rises. The effect of slag basicity ($R = w(\mathrm{CaO})/w(\mathrm{SiO_2})$, 2~10) shows a similar tendency. The TiO$_2$ content and slag basicity evidently affect the precipitated phases in the slags at a lower temperature (e.g., 1310 °C). With the increase in basicity, the liquid areal fraction increases first, and then decreases. Moreover, the CaO-TiO$_x$-Al$_2$O$_3$ phase (CTA) and its TiO$_x$ content show a declining trend at 1310 °C. When $R = 10$, large amounts of solid calcium aluminates are precipitated. With TiO$_2$ addition in the slags, the TiO$_x$ contents in both liquid and CTA phases increase. Excessive TiO$_2$ addition (e.g., 10%) leads to the large precipitation of CTA, as well. To improve the melting properties of the slag and the yield of Ti alloys during the refinement of Ti-bearing steel grades, a small TiO$_2$ addition (e.g., 5%) may be considered.

**Keywords:** TiO$_2$; refining slag; melting temperature; basicity; phase precipitation





## 1. Introduction

Ladle refining slag plays a vital role in the production of clean steel grades. A suitable refining slag system generally requires good physical and chemical properties, aiming at efficient desulfurization, deoxidation, and inclusion absorption, as well as inclusion modification [1–3].

The quaternary slag system of CaO-MgO-SiO$_2$-Al$_2$O$_3$ is widely used as a secondary refining slag. In the case of Ti-bearing steel grades, this slag system is also chosen. In real industrial practice, the Ti content in Ti-bearing steel grades is generally in the range of 0.015% to 0.22%. After the addition of Ti alloys in liquid steel, some of the dissolved Ti in liquid steel will transfer to the top refining slag due to the slag-steel equilibrium, forming a CaO-SiO$_2$-Al$_2$O$_3$-MgO-TiO$_x$ system slag. The generation of TiO$_x$ in slag will not only affect the physical and chemical properties of the slag, but also increase the loss of Ti alloys. In fact, to weaken the Ti loss, some TiO$_x$ can be added to the slag before refinement. Considering the potential influence of TiO$_x$ on the metallurgical performance of the slag, many researchers [4–7] carried out a series of investigations to understand its effect.

Due to the utilization of vanadium titanomagnetite, there are many studies focusing on TiO$_2$-containing blast furnace (BF) slags. A large amount of researchers [8–27] believed that TiO$_2$ evidently affects the physical and chemical properties of BF slags, e.g., viscosity [8–14] and melting point [15–27]. Several studies [8–13] investigated the effect of TiO$_2$ on the viscosity of CaO-SiO$_2$-Al$_2$O$_3$-MgO slag system, and indicated that TiO$_2$ depolymerized the structure of silicates and decreased slag viscosity. At the same time, Yan et al. [13] pointed

out that excessive $TiO_2$ would lead to the precipitation of perovskite ($CaTiO_3$) in the slag, resulting in an increase in viscosity. Feng et al. [14] found that the sum of pyroxene and perovskite phases in the slag increased with the $TiO_2$ content, as well.

On the other hand, the liquidus temperature and phase equilibrium of $TiO_2$-containing slag system were the focus of some investigators [15–27]. With the increase in $TiO_2$ content in BF slags, the variation trend of liquidus temperature was different in different references. Some publications [15–20] showed a decrease in liquidus, while it increased first, and then dropped in the study of Gao et al. [21]. In contrast, Osborn et al. [23] believed that the liquidus was not clearly affected by $TiO_2$. In addition, in the slag composition ranges of Zhen et al. [24], a liquid region and two solid–liquid coexistence regions (liquid-$CaTiO_3$ and liquid-$TiO_2$) were obtained at 1500 °C. Moreover, Shi et al. [25] calculated the liquidus of the $CaO-SiO_2-5\%MgO-10\%Al_2O_3-TiO_2$ slag system, and found that the liquid phase and ($C_2MS_2$, $C_2AS$)$_{ss}$ solid solution, as well as the $CaTiO_3$ phase coexisted in the slag at 1300~1500 °C. Jiao et al. [26] reported that when the $TiO_2$ content increased from 5% to 15%, the melting point of experimental BF slags increased, and the primary phase was $CaTiO_3$. Except for BF slags, Wang et al. [27] found that the melting point of $CaO-SiO_2-MgO-Al_2O_3$ electroslag system increased with the $TiO_2$ content (48~54%), and the higher slag basicity was favorable for the generation of $CaTiO_3$ in the slag.

It is well known that BF slags contain a high content of $SiO_2$ (low basicity), and generally, the $TiO_2$ content in BF slag is in the range of 10~47% [15] when vanadium titanomagnetite is applied. In fact, ladle slags have significantly higher basicity, and the $TiO_2$ in ladle slags cannot reach that content in BF slags. Although there are many studies on the effect of $TiO_2$ on BF slags as mentioned above, the effect of $TiO_2$ on refining slags is still not clear due to the different composition ranges. Therefore, further studies need to reveal this effect.

In the present study, some laboratory experiments were carried out to investigate the effect of $TiO_2$ addition on the melting behaviors of $CaO-MgO-SiO_2-Al_2O_3$ refining slags. According to the measurement of the melting points and the analysis of the phases in the slags at different temperatures, the effects of $TiO_2$ and slag basicity on the melting behaviors were discussed. Moreover, the possible additional amount of $TiO_2$ was suggested based on the experimental results.

## 2. Materials and Methods

### 2.1. Slag Preparation

Chemical reagents of CaO, $SiO_2$, $Al_2O_3$, MgO, and $TiO_2$ (Sinopharm Chemical Reagent, Shanghai, China) were used to prepare the refining slags. Both mixed and pre-melted slags were considered in the present study. When mixed slags were employed, the chemical powders were well-mixed based on the compositions presented in Table 1. To obtain the phases in slag at high temperatures, pre-melted slags were prepared using an electric resistance furnace as shown in Figure 1.

**Table 1.** Compositions (mass%) and measured melting points ($T_m$, °C).

| Slag No. | CaO | SiO₂ | Al₂O₃ | MgO | TiO₂ | Basicity $R$ | Heating Temp (°C) | Measured $T_m$ (°C) | |
|---|---|---|---|---|---|---|---|---|---|
| | | | | | | | | Mixed | Melted |
| A1 | 55.80 | 9.20 | 30.00 | 5.00 | - | 6 | 1550, 1310 | 1313.6 | 1286.4 |
| A2 | 54.00 | 9.00 | 30.00 | 5.00 | 2.00 | 6 | 1550 | 1295.1 | 1282.4 |
| A3 | 51.43 | 8.57 | 30.00 | 5.00 | 5.00 | 6 | 1550, 1310 | 1293.1 | 1280.8 |
| A4 | 49.71 | 8.29 | 30.00 | 5.00 | 7.00 | 6 | - | 1294.2 | - |
| A5 | 47.14 | 7.86 | 30.00 | 5.00 | 10.00 | 6 | 1550, 1310 | 1301.4 | 1309.6 |
| B1 | 40.00 | 20.00 | 30.00 | 5.00 | 5.00 | 2 | 1550, 1310 | 1418.0 | 1342.8 |
| B2 | 53.33 | 6.67 | 30.00 | 5.00 | 5.00 | 8 | 1550 | 1299.1 | 1298.7 |
| B3 | 54.54 | 5.45 | 30.00 | 5.00 | 5.00 | 10 | 1550, 1310 | 1325.0 | 1305.2 |

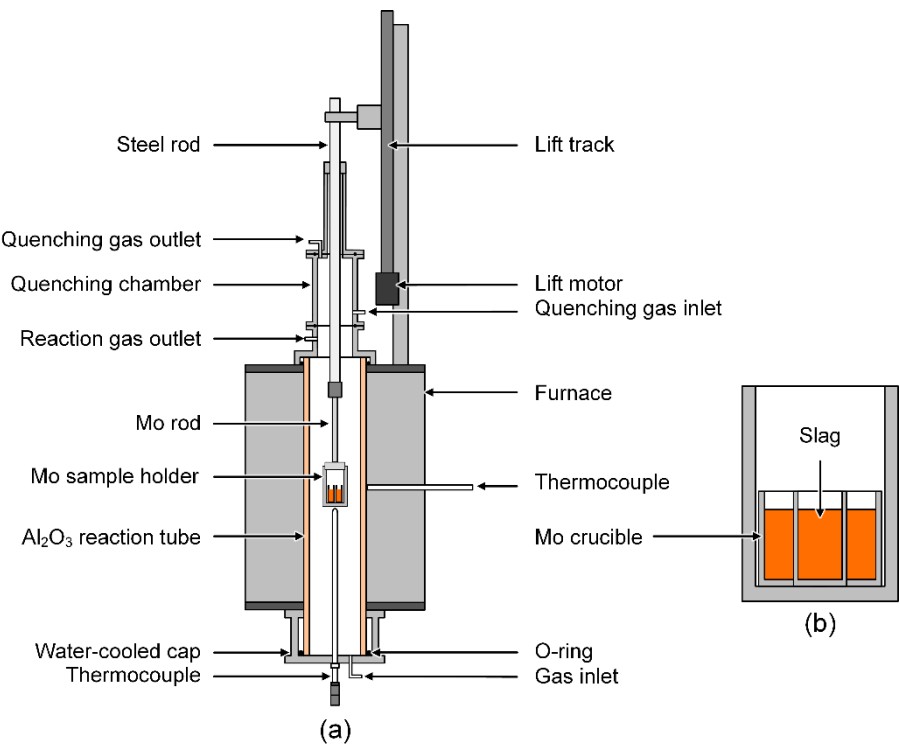

**Figure 1.** Experimental setup: (**a**) Furnace; (**b**) sample holder with crucibles.

The experimental furnace mainly includes an $Al_2O_3$ reaction tube and a water-cooled quenching chamber, which are sealed by some O-rings. Two B-type thermocouples were employed to control and measure the temperatures of the reaction tube and the samples inside. Before the experiment, 10 g of well-mixed chemical powders were placed in a molybdenum crucible (OD17 mm, ID20 mm, H55 mm). As shown in Figure 1, two or three molybdenum crucibles considered as one set were placed in a molybdenum sample holder, and then lowered down to the hot zone of the furnace by a motor-driven suspension steel rod. Thereafter, the furnace was completely sealed and evacuated by a vacuum pump. After evacuation, high-purity argon (>99.999%, Shuntai Special Gas, Shenyang, China) was introduced into the furnace, and finally maintained a flow rate of 0.2 NL/min. Thereafter, the furnace was heated, and when the sample temperature reached 1550 °C, it remained for 1 h to allow the slag to fully melt. For phase observation at a temperature around the melting point, the temperature of the samples was then dropped to 1310 °C and remained for another hour to allow the solid phase to precipitate. The detailed melting temperatures of each slag are listed in Table 1. Next, the sample holder was rapidly raised by the steel rod to the quenching chamber, and argon gas with a high flow rate was injected to accelerate quenching at the same time.

*2.2. Measurement of Melting Point*

The melting points of the slags were measured by the RDS-05 measuring device (Northeastern University, Shenyang, China), which is schematically shown in Figure 2. This device mainly consisted of a PID-controlled furnace and a digital camera. Before the measurement, the slags were ground into fine powders, and then pressed in a tiny cylinder ($\phi3 \times 3$ mm) by a mold. When measured, the pressed slag cylinder was placed on a Pt sheet, and then on the surface of an $Al_2O_3$ plate as shown in Figure 2. The $Al_2O_3$ plate was driven by a motor to the hot zone of the furnace, and then the furnace was heated at a rate of 10 °C/min. The camera was fixed to capture the height variation of the slag, and the photos at different temperatures were saved by a computer. When the height declined by 50%, the corresponding temperature was defined as the hemispherical melting point ($T_m$). Figure 3 shows an example of the photos of a sample at different melting stages. For each

slag, the measurement was repeated three times, and the average value was considered as the melting point in this study.

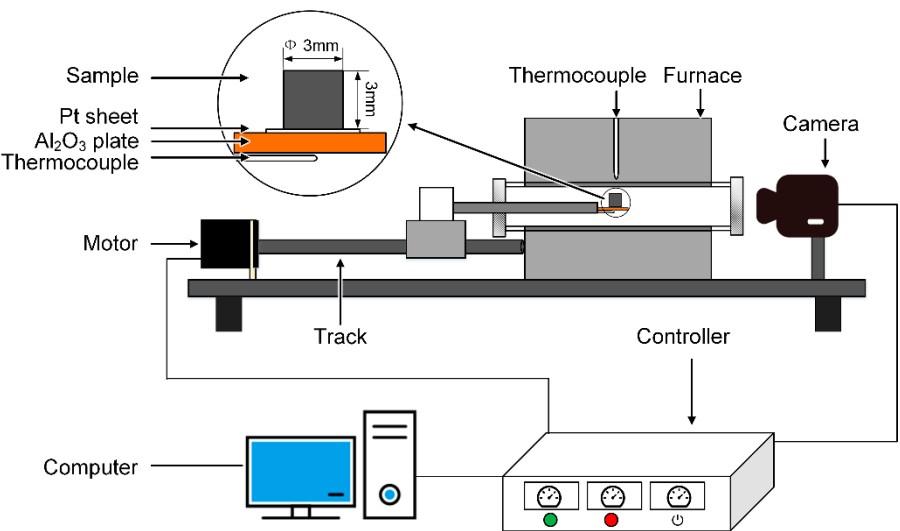

**Figure 2.** Illustration of RDS-05 measuring device for the slag melting point.

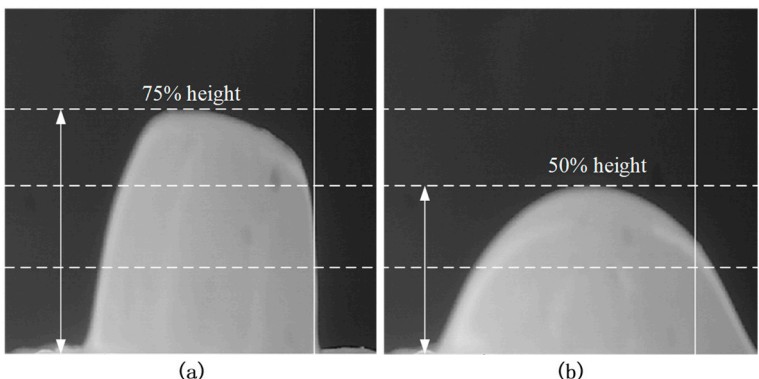

**Figure 3.** Photos of a sample at different melting stages: (**a**) 75% height; (**b**) 50% height.

### 2.3. Sample Analysis

The quenched slags were cut and polished for observation. The morphologies, compositions, and element distributions of the phases in the slags were analyzed by a scanning electron microscope (SEM, Zeiss EVO 18, Carl Zeiss, Jena, Germany, accelerating voltage 20 kV) and attached energy dispersive spectroscopy (EDS). Additionally, mage analysis was considered to evaluate the fraction of the liquid phase, although it has some uncertainties.

## 3. Results

### 3.1. Variation of Melting Point

#### 3.1.1. Effect of Slag Basicity

The measured melting points of the mixed slag and pre-melted slags are listed in Table 1. For easy comparison, Figure 4 presents the changing trend of melting point of the slags ($w(TiO_2)$ = 5%) with different basicity levels ($R = w(CaO)/w(SiO_2)$). It can be seen from Figure 4 that the melting points of pre-melted slags are generally lower than the mixed slags. This gap is very likely due to the melting procedure of the slags. After pre-melting, some low melting point phases were already formed in the slags. In contrast, the powders of CaO, MgO, $Al_2O_3$, and $SiO_2$ in the mixed slags all have high melting points before the measurement. During the measurement, the pre-melted slags can form a liquid phase and melt more easily, and apparently show a lower melting point. Similar results were also obtained in the case of BOF slag in the study of Yan et al. [28]. On the other hand,

the melting point changing trend of the pre-melted and mixed slags is well-consistent. With the increase in slag basicity, the melting points of the slags first show a sharp decline tendency, and then start to rise slightly. When the basicity of the $TiO_2$-containing slag is 6, its melting point reaches the minimum (1293.1 °C).

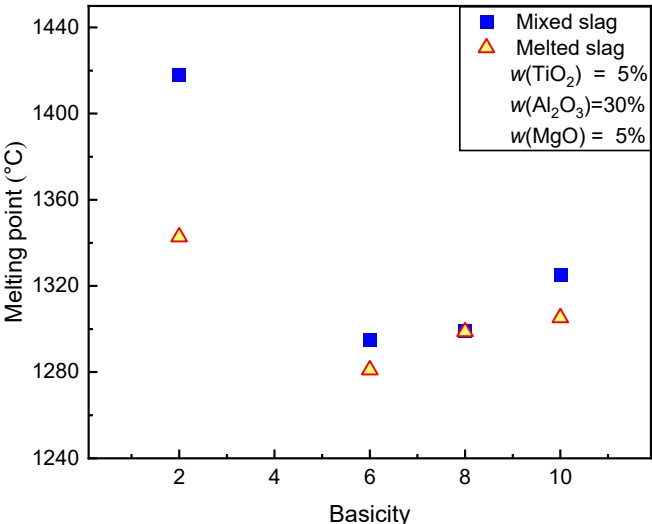

**Figure 4.** Effect of slag basicity on the melting points of slags.

### 3.1.2. Effect of $TiO_2$ Addition

The effect of $TiO_2$ addition on the melting points of the refining slags ($R$ = 6) is also plotted in Figure 5. Similarly, the difference between the pre-melted slags and mixed slags can be found in Figure 5. As shown in Figure 5, the melting points of the slags decrease first, and then increase with the addition of $TiO_2$ in the slags. When the $TiO_2$ content is 5%, it obtains the lowest melting point (1293.1 °C). Jiao et al. [26] measured the melting temperature of a BF slag ($CaO$-$SiO_2$-11.26%$Al_2O_3$-9.13%$MgO$-15%$FeO$-$TiO_2$, $R$ = 1.32). For comparison, their data are plotted in Figure 5. As can be seen from Figure 5, when the $TiO_2$ content increases from 5% to 10% in the BF slag, the melting temperature increases from 1230 to 1232 °C. Due to the difference in slag compositions, the measured values are higher in this study.

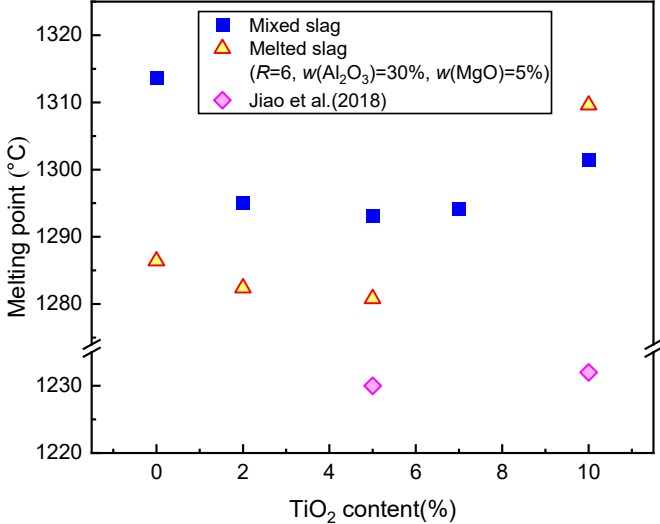

**Figure 5.** Effect of $TiO_2$ content on the melting points of slags. Data by Jiao et al. refer to Ref. [26].

*3.2. Phases in Slag*

3.2.1. Effect of Basicity

Figure 6 shows the morphologies of Slags A1 and B1 at 1550 °C. It can be seen that only the liquid phase is presented in the slags. Although Figure 6 only presents two images of the slags, it was confirmed by SEM that only a solid amorphous structure exists in the slags, indicating that they are in the liquid phase at 1550 °C. In addition, Slags A1 and B1 have the highest melting points as presented in Table 1, and this proves that these slags should be in the liquid phase. In the industrial practice, the refining temperature can reach 1600 °C or even higher, indicating that all the experimental slags should be completely melted at the refining temperature. On the other hand, the top layer of the refining slag is in contact with the air, and the temperature is significantly lower than the contact with liquid steel. In this case, if the slag has a high melting point, the top layer may form a slag crust and influence the refining efficiency. To understand the melting behaviors of the slags comprehensively, the temperature near the melting point is further considered to observe the phases.

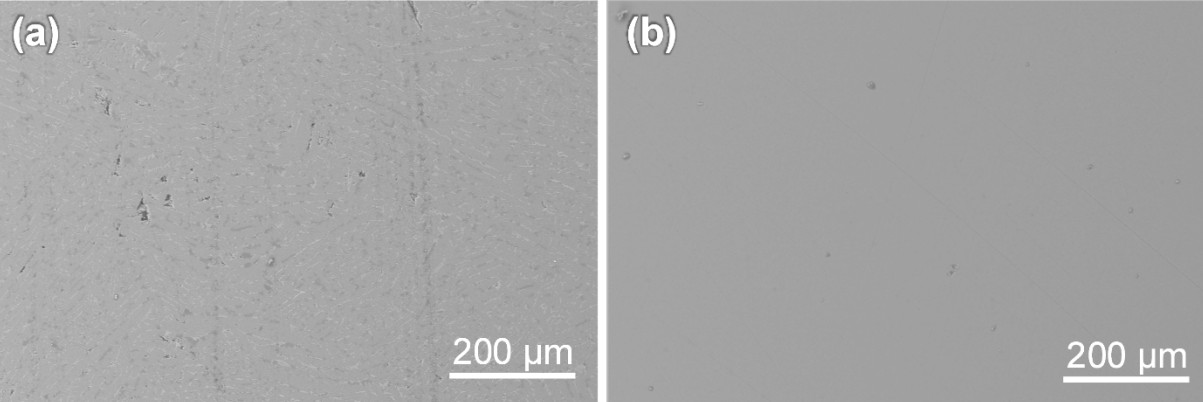

**Figure 6.** Morphologies of slag samples at 1550 °C: (**a**) Slag B1, $R = 2$, $w(TiO_2) = 5\%$; (**b**) Slag A1, $R = 6$, $w(TiO_2) = 0\%$.

The morphologies of Slags A3, B1, and B3 at 1310 °C are shown in Figure 7. It is evidently seen that both the liquid (e.g., Points 4, 5, and 12) and solid phases are generated in these slags. The EDS compositions of these phases are listed in Table 2. As shown in Table 2, the compositions of the solid phases vary in different slags.

When the slag basicity is 2 (Slag B1, $w(TiO_2) = 5\%$), three solid phases are shown in Figure 7a. Combined with the composition in Table 2, $2CaO \cdot Al_2O_3 \cdot SiO_2$ ($C_2AS$, gray phase, e.g., Point 1), $MgO \cdot Al_2O_3$ spinel (MA, dark gray phase, e.g., Point 2), and $CaO\text{-}TiO_x\text{-}Al_2O_3$ (CTA, light gray phase, e.g., Point 3) are found as the main solid phases in Slag B1. It is noted that the color of $C_2AS$ phase is very close to the liquid phase (e.g., Point 4). For more details on the slag, Figure 8 presents the elemental mappings of the slag at 1310 °C. In Figure 8, the three solid phases are clearly seen, and only a small amount of liquid phase is distributed among the solids.

Figure 7b presents the morphology of Slag A3 ($R = 6$, $w(TiO_2) = 5\%$) at 1310 °C. Similarly, there are three solid phases in this slag. Figure 7b and the composition provided in Table 2 indicate that the solid phases are CTA (light gray phase, e.g., Point 6) and $2CaO \cdot SiO_2$ ($C_2S$, gray phase, e.g., Point 7), as well as some MgO islands (black phase, e.g., Point 8).

In the case of Slag B3 ($R = 6$, $w(TiO_2) = 5\%$) shown in Figure 7c, four solid phases were detected at 1310 °C, namely, MgO (e.g., Point 9), calcium aluminate ($3CaO \cdot Al_2O_3$, $C_3A$, e.g., Point 10), and CTA (e.g., Point 11), as well as $C_2S$ (e.g., Point 13). The MgO islands are in black color. The colors of $C_3A$, CTA, and $C_2S$ are similar, and they need to be distinguished by EDS.

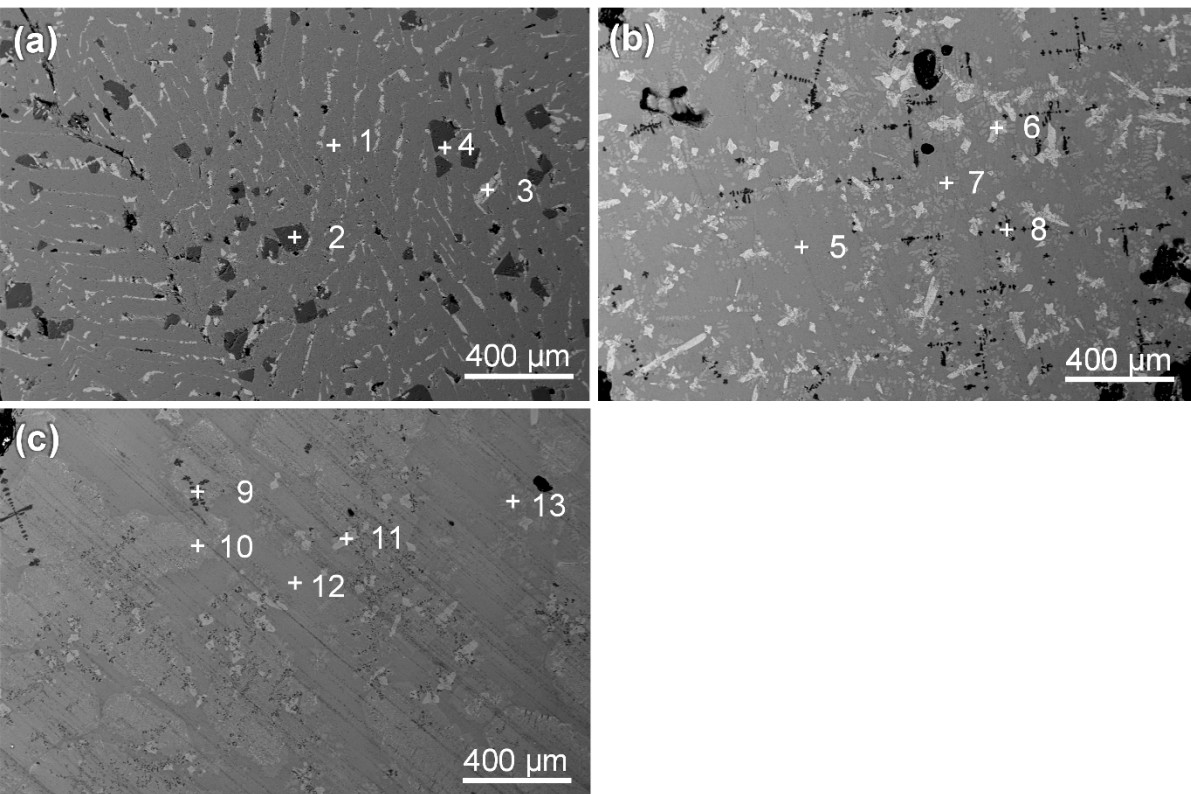

**Figure 7.** Morphologies of slag samples with different basicity levels ($w(TiO_2)$ = 5%, 1310 °C): (**a**) Slag B1, $R$ = 2; (**b**) Slag A3, $R$ = 6; (**c**) Slag B3, $R$ = 10.

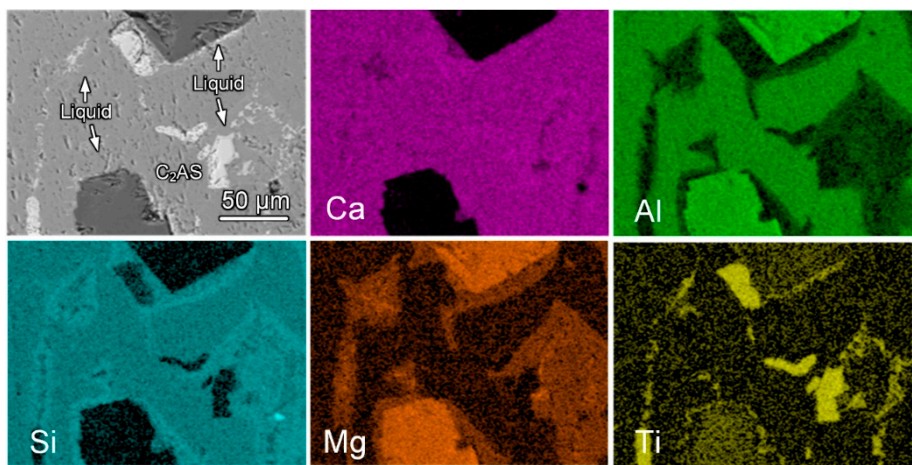

**Figure 8.** Elemental mappings of Slag B1 at 1310 °C ($R$ = 2, $w(TiO_2)$ = 5%).

As shown in Figure 7, with the increase in slag basicity, the liquid areal fraction increases first, and then decreases. Moreover, the amount of the CTA phase shows a declining trend. Moreover, the measured liquid ratio by the image analysis supports this phenomenon. When the basicity increases from 2 to 6, the ratio of liquid phase increases from 0.51% to 30.28%. When the basicity further increases to 10, the liquid phase ratio drops to 17.69%. At the same time, the average atomic fractions of Ti element in the liquid and CTA phases of the slags are plotted with the change in basicity (Figure 9). It is found that the $TiO_x$ content in the liquid phase increases, while it declines in the CTA phase with the increase in slag basicity. In fact, the changing trend can be found in Table 2.

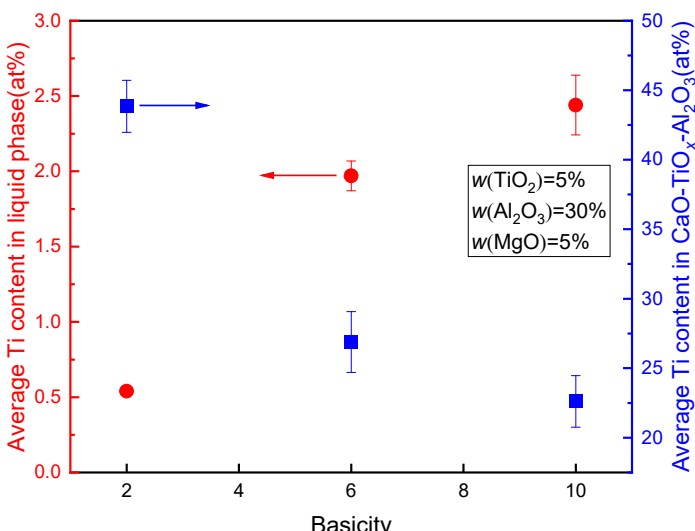

**Figure 9.** Ti elemental content in the liquid and CTA phases with different slag basicity levels.

**Table 2.** EDS results of different points in Figures 7 and 10 (at%).

| Point | Figure | Slag | Si | Ca | Al | Mg | Ti | O | Estimated Phase |
|-------|--------|------|------|-------|-------|-------|-------|-------|-----------------|
| 1 | 7a | B1 | 9.87 | 20.26 | 18.10 | 0.51 | 0.27 | 50.98 | $2CaO \cdot Al_2O_3 \cdot SiO_2$ |
| 2 | 7a | B1 | 0.06 | - | 26.70 | 15.01 | 0.60 | 57.63 | $MgO \cdot Al_2O_3$ |
| 3 | 7a | B1 | 0.09 | 22.02 | 1.71 | 0.08 | 19.69 | 56.41 | $CaO\text{-}TiO_x\text{-}Al_2O_3$ |
| 4 | 7a | B1 | 14.48 | 18.44 | 4.54 | 6.13 | 0.18 | 56.22 | Liquid |
| 5 | 7b | A3 | 2.79 | 22.57 | 17.92 | 2.95 | 0.90 | 52.87 | Liquid |
| 6 | 7b | A3 | 1.15 | 23.09 | 6.73 | 1.24 | 12.80 | 55.00 | $CaO\text{-}TiO_x\text{-}Al_2O_3$ |
| 7 | 7b | A3 | 12.59 | 27.52 | 1.75 | 0.41 | 0.65 | 57.08 | $2CaO \cdot SiO_2$ |
| 8 | 7b | A3 | 0.04 | 0.73 | 1.56 | 50.48 | 0.02 | 47.17 | MgO |
| 9 | 7c | B3 | 0.03 | 0.27 | 0.11 | 42.55 | 0.05 | 56.99 | MgO |
| 10 | 7c | B3 | 2.09 | 28.54 | 15.13 | 0.95 | 1.16 | 52.14 | $3CaO \cdot Al_2O_3$ |
| 11 | 7c | B3 | 0.92 | 24.22 | 8.83 | 1.27 | 10.30 | 54.46 | $CaO\text{-}TiO_x\text{-}Al_2O_3$ |
| 12 | 7c | B3 | 2.49 | 25.67 | 19.82 | 3.23 | 1.35 | 47.43 | Liquid |
| 13 | 7c | B3 | 11.16 | 24.32 | 2.05 | 0.98 | 1.24 | 60.24 | $2CaO \cdot SiO_2$ |
| 14 | 10a | A1 | 2.06 | 20.27 | 21.59 | 3.77 | 0.10 | 52.22 | Liquid |
| 15 | 10a | A1 | 0.14 | 0.31 | 0.08 | 52.54 | - | 46.93 | MgO |
| 16 | 10a | A1 | 13.68 | 25.38 | 1.71 | 0.45 | - | 58.78 | $2CaO \cdot SiO_2$ |
| 17 | 10b | A5 | 2.97 | 21.65 | 19.53 | 3.27 | 0.79 | 51.80 | Liquid |
| 18 | 10b | A5 | 0.92 | 24.22 | 8.83 | 1.27 | 10.3 | 54.46 | $CaO\text{-}TiO_x\text{-}Al_2O_3$ |
| 19 | 10b | A5 | 12.69 | 25.87 | 0.80 | 0.95 | 0.34 | 59.34 | $2CaO \cdot SiO_2$ |
| 20 | 10b | A5 | 0.09 | 2.96 | 1.02 | 51.35 | 0.07 | 44.51 | MgO |

### 3.2.2. Effect of TiO$_2$ Addition

The morphologies of the slags with different $TiO_2$ additions are presented in Figure 10. At 1310 °C, both the liquid and solid phases are shown in the slags. The EDS compositions of these phases are also listed in Table 2. Combined with Figure 7b ($w(TiO_2) = 5\%$), it can be seen from Figure 10 that with the increase in $TiO_2$ content in slag, the fraction of the liquid phase increases first, and then drops. According to the measured liquid ratio in the slags, it is found that when the addition of $TiO_2$ increases from 0 to 5%, the liquid ratio increases from 27.48% to 30.28%, and further adding $TiO_2$ to 10% leads to the decline in the ratio to 26.29%. The change in liquid ratio holds the key to the variation of the melting point. As shown in Figure 10a, MgO (in black color, e.g., Point 15) and C$_2$S (in light gray color, e.g., Point 16) are the solid phases in the $TiO_2$-free slag (Slag A1, $R = 6$) at 1310 °C, and the

amount of $C_2S$ particles is remarkable due to the small liquid ratio (27.48%) in the slag. When the $TiO_2$ content is 10% (Slag A5, $R = 6$), the solid phases are the same as in Figure 7b. The amount of CTA (e.g., Point 18) becomes more profound, and the $C_2S$ particles (e.g., Point 19) are evidently reduced (Figure 10b).

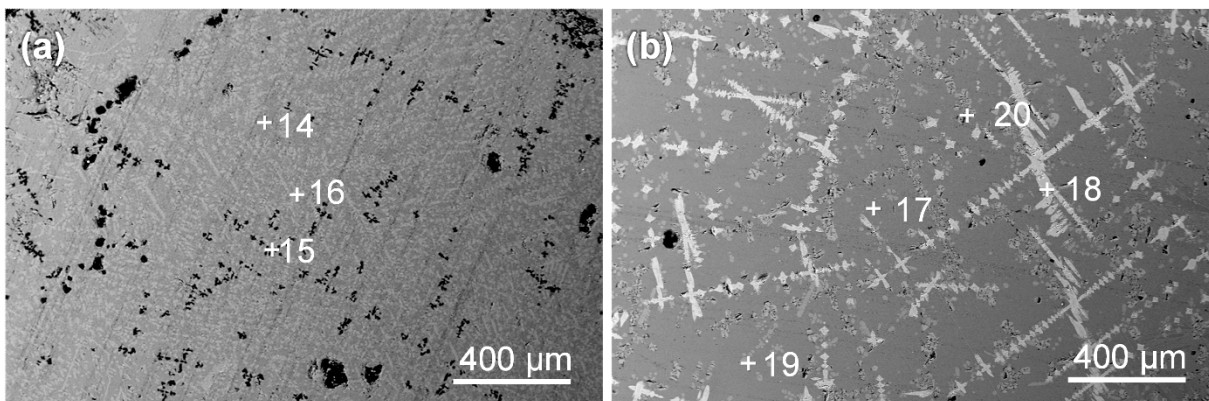

**Figure 10.** Morphologies of slags with different $TiO_2$ contents ($R = 6$, 1310 °C): (**a**) Slag A1, $w(TiO_2) = 0\%$; (**b**) Slag A5, $w(TiO_2) = 10\%$.

To obtain the distribution of Ti element in the slags, Figure 11 shows the elemental mappings of Slags A1 and A3. It can be seen from Figure 11b that the Ti element in these slags is not only distributed in the liquid phase, but also in the CTA solid phase. The content of Ti element in the liquid phase is evidently lower than the CTA solid phase. Additionally, the solid phases of $C_2S$ and MgO are clearly presented in Figure 11.

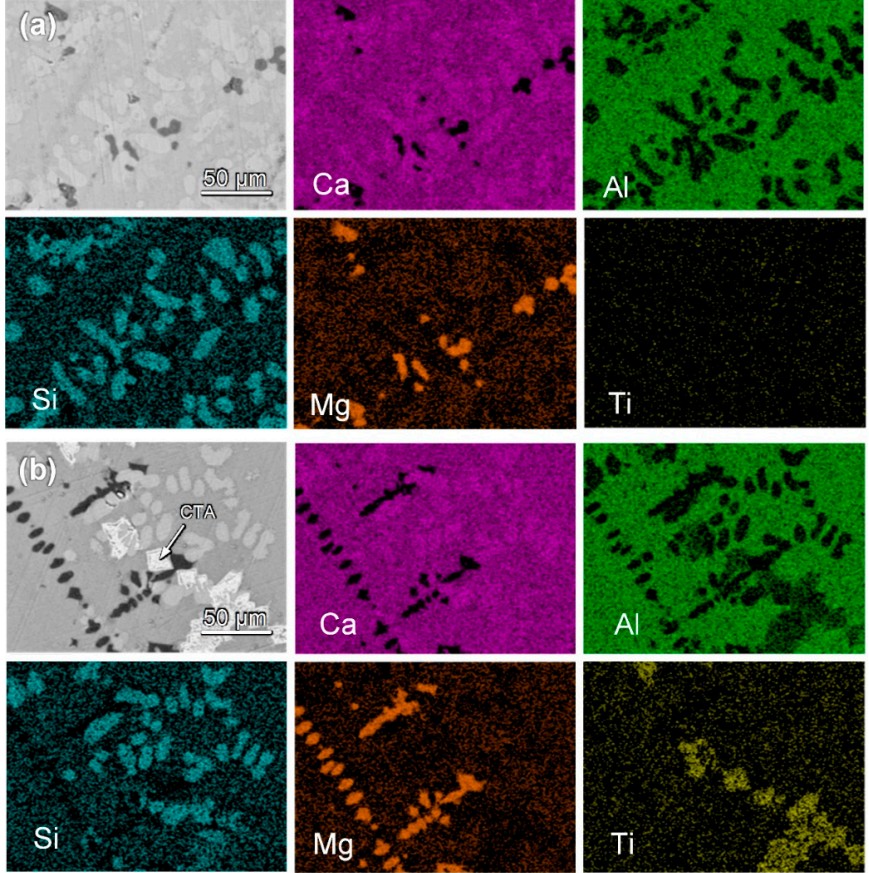

**Figure 11.** Elemental mappings of slag with different $TiO_2$ contents (1310 °C, $R = 6$): (**a**) Slag A1, $w(TiO_2) = 0\%$; (**b**) Slag A3, $w(TiO_2) = 5\%$.

Figure 12 further plots the average atomic fractions of Ti element in the liquid and CTA phases of these slags with $TiO_2$ addition. It is shown that with the increase in $TiO_2$ content in the slags, the $TiO_x$ contents in both liquid and CTA phases increase, as well. Of note, when the $TiO_2$ content is higher than 5%, the increasing trend becomes very weak in the liquid phase. In contrast, the $TiO_x$ content increases evidently in the CTA phase.

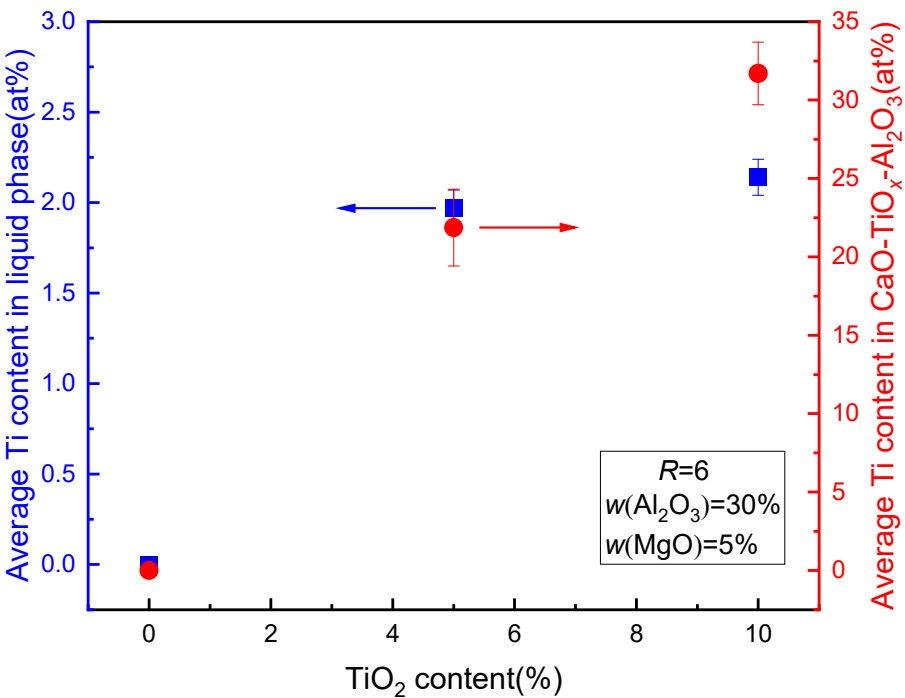

**Figure 12.** Ti elemental content in the liquid and CTA phases with different $TiO_2$ contents.

## 4. Discussion

### 4.1. Effect of Basicity on Melting Behaviors

It can be seen from Figure 4 that when the basicity of slag increases, the melting points of the $TiO_2$-containing slags ($w(TiO_2)$ = 5%) decrease first, and then increase. As shown in Figure 7, the liquid and solid phases in these slags change accordingly at 1310 °C. This implies that the effect of the slag basicity on the melting points of the slags is mainly based on the phase formation during melting.

As mentioned above, when the slag basicity is very low, e.g., $R$ = 2 (Slag B1), the solid phases of $C_2AS$, MA spinel, and CTA are found (Figures 7a and 8). In the study of Shi et al. [23], the phases of $C_2AS$ and $CaO \cdot TiO_2$ (CT) were also detected in the low basicity $TiO_2$-containing slag at 1300~1500 °C. To present the effect of CTA phase in this study, Figure 13 illustrates the phase diagram of $CaO$-$Al_2O_3$-$TiO_2$ system [29]. The detailed compositions of the CTA phase presented in Table 2 are also marked in this figure. It can be seen from Figure 13 that the composition of CTA changes along the connection line between CT and $C_3A$. In the case of Slag B1, the content of $Al_2O_3$ in CTA is very low (lower than 5%, Table 2), and its melting point is close to the CT ($T_m$ = 1960 °C [30]). Meanwhile, it is reported that the melting points of $C_2AS$ and MA spinel are 1590 [31] and 2135 °C [30], respectively. Due to the low liquid fraction (Figure 8) and these high melting point phases, the melting point of Slag B1 is the highest as shown in Figure 4.

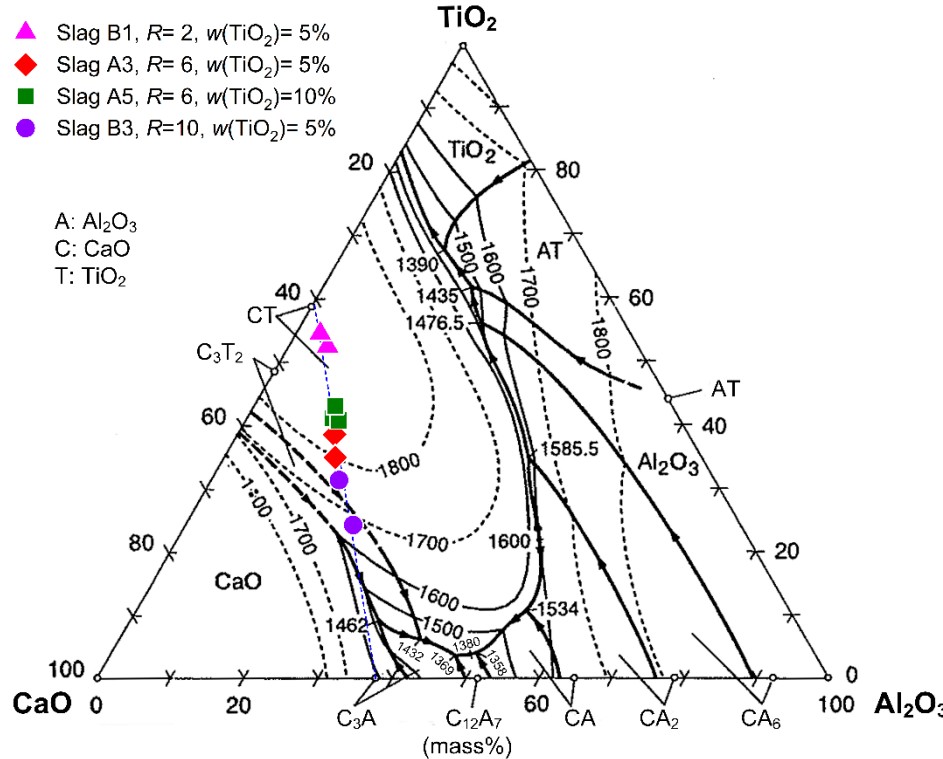

**Figure 13.** CTA composition plotted in CaO-Al$_2$O$_3$-TiO$_2$ liquidus projection. Adapted from Ref. [29].

With the increase in slag basicity, large amounts of C$_2$AS and MA spinel phase disappear completely, and a few MgO and C$_2$S islands are detected (Figure 7b,c). At the same time, the liquid phase is evidently enlarged from 0.51% to 30.28%. Additionally, as shown in Figure 9, the content of TiO$_x$ decreases in the CTA phase and increases in the liquid phase. According to Figure 13, the melting point of the CTA phase should be declined. Therefore, Slag A3 ($R$ = 6) has a lower melting point in contrast to Slag B1 ($R$ = 2). When the basicity of slag further increases, e.g., $R$ = 10 (Slag B3), large amounts of solid C$_3$A ($T_m$ = 1539 °C [32]) are precipitated as shown in Figure 7c. Although the fractions of the CTA phase and C$_2$S drop in Figure 7c, the decreasing liquid phase (from 30.28% to 17.69%) and the large C$_3$A precipitation can still lead to an increase in the melting point. Due to the precipitation of solid phase and the variation of liquid phase, the melting point of the slags changes accordingly.

Notably, 5% of MgO and 5% of TiO$_2$ are contained in these slags. To compare the difference between the TiO$_2$-containing and TiO$_2$-free slags, the compositions of Slags A3 and B1-B3 are also simply plotted in the CaO-SiO$_2$-Al$_2$O$_3$ ternary phase diagram [29] as shown in Figure 14. Moreover, it can be seen from Figure 14 that the liquidus temperature of the plotted slags increases first, and then decreases. This changing tendency is in line with the experimental TiO$_2$-containing slags. In fact, note that the measured values are hemispherical melting points ($T_m$), and they are not absolutely equal to the liquidus temperature. Therefore, some deviations can still be found between the phase diagram and the measured values shown in Table 1. In fact, similar measurements [27,28,31,33] were also conducted to investigate the slag melting behaviors. Furthermore, the trend shown in Figure 14 indicates that a small additional amount of MgO and TiO$_2$ does not evidently influence the impact of basicity on this slag system. This indicates that a suitable addition of TiO$_2$ may not change the refining effect of the slag. Nevertheless, this merits further investigations in the near future.

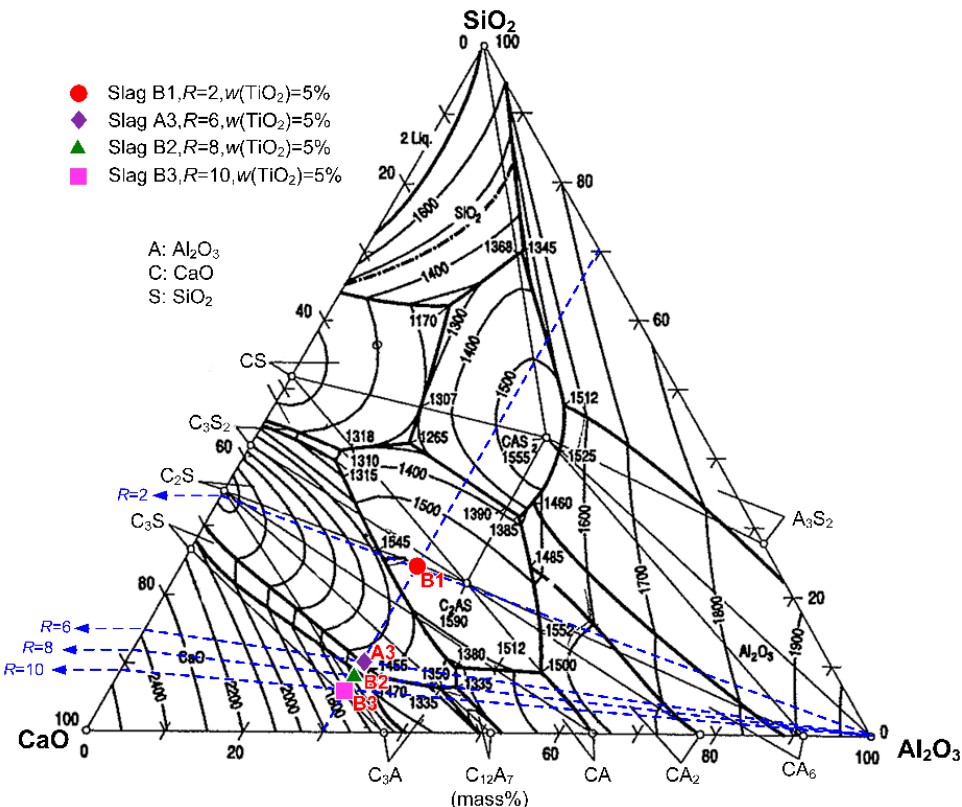

**Figure 14.** Slag composition plotted in CaO-SiO$_2$-Al$_2$O$_3$ liquidus projection. Adapted from Ref. [29].

### 4.2. Effect of TiO$_2$ on Melting Behaviors

As shown in Figure 5 and Table 1, the melting points of the slags ($R$ = 6) decrease first, and then increase with TiO$_2$ addition. Meanwhile, according to the morphologies of the slags (Figures 7b and 10) and the TiO$_x$ contents in both liquid and CAT phases (Figure 12 and Table 2), it can be concluded that in addition to slag basicity, the TiO$_2$ content in the slag influences its melting behaviors.

It can be seen from Figure 10a and Table 2 that there are evident C$_2$S particles precipitated in the TiO$_2$-free slag (Slag A1) at 1310 °C. After TiO$_2$ addition in Slag A2, the liquid phase increases from 27.48% to 30.28%, and some TiO$_x$ is dissolved in the liquid phase as shown in Figure 12. Additionally, the fraction of C$_2$S particles is significantly reduced, and the CTA phase is observed as shown in Figures 7b and 10b. This indicates that the addition of TiO$_2$ not only influences the generation of liquid phase, but also results in the transformation of C$_2$S and CTA. As reported in [31], the melting point of C$_2$S is 2130 °C, while the melting point of CTA in Slag A3 is estimated to be lower than 1800 °C according to Figure 13. Therefore, moderate TiO$_2$ addition at the beginning could reduce the melting points of the slags.

When the TiO$_2$ content is higher than 5%, the melting point of the slags starts to rise as shown in Figure 5. From the phases in Figures 8 and 11b, as well as Table 2, it is found that the Ti element in the TiO$_2$-containing slags is mainly distributed in the CTA phase at 1310 °C, while its content is quite low (<3%) in the liquid phase. As shown in Figure 12, with further addition of TiO$_2$ (higher than 5%), the TiO$_x$ content in the CTA phase increases evidently, while the TiO$_x$ content in the liquid phase changes very slightly. In this case, the TiO$_x$ in the liquid phase seems to be saturated at 1310 °C; therefore, further addition of TiO$_2$ would result in more CTA precipitated, e.g., Figure 10b. On the other hand, it can be seen from Figure 13 that the melting point of precipitated CTA increases with the TiO$_x$ content. Consequently, the liquid phase will decrease from 30.28% to 26.29%, resulting in the rise in the melting point.

In Refs. [15–27], the effect of $TiO_2$ on the melting behaviors of BF slags showed different trends as described in the introduction part. Many researchers pointed out that $TiO_2$ would lead to the rise in the melting point of BOF slags due to the high content of $TiO_2$. In contrast, when a small amount of $TiO_2$ is added to the refining slags, the melting property of the slags is improved, in contrast to the $TiO_2$-free slag, although it increases when $w(TiO_2) \geq 5\%$ as shown in Figure 10. This further confirms that the effect of $TiO_2$ on the refining slags differs from the BF slags.

As discussed above, a small amount of $TiO_2$ does not clearly affect the role of basicity on the melting behaviors of $CaO$-$SiO_2$-$Al_2O_3$-$MgO$ slags. Meanwhile, when the basicity is the same, the addition of $TiO_2$ still influences the melting behaviors. According to the present study, it is found that a small amount of $TiO_2$ can reduce the melting point of the slag, while the excessive addition of $TiO_2$ will lead to the large precipitation of CTA at a low temperature. These solid precipitates influence not only the melting of the slag, but also its viscosity. Due to the potential to improve the melting of the slag and the yield of Ti alloys, $TiO_2$ addition (e.g., 5%) may be considered for the refining of Ti-bearing steel grades in industry. Nevertheless, a few other factors, e.g., deoxidation, desulfurization, and inclusions, etc., should be taken into account as well, and future studies are required to clarify the effect of $TiO_2$ addition on these factors.

## 5. Conclusions

According to the melting point measurement and phase analysis, the melting behaviors of $CaO$-$SiO_2$-5%$MgO$-30%$Al_2O_3$-$TiO_2$ refining slags were studied, and the main conclusions are summarized as follows:

(1) With the increase in $TiO_2$ content in slag, the melting point of the slags drops first, and then rises. A lower melting point is obtained when the $TiO_2$ content is around 5%. The effect of slag basicity shows a similar tendency, and its effect is not clearly influenced by a small amount of $TiO_2$.

(2) The $TiO_2$ content and slag basicity evidently affect the precipitated phases in the slags at a lower temperature. With the increase in basicity, the liquid areal fraction at 1310 °C increases first, and then decreases. Moreover, the CTA phase and $TiO_x$ content in this phase show a declining trend. When the basicity is 10, a large number of solid calcium aluminates are precipitated in the slag. With the addition of $TiO_2$ in the slags, the $TiO_x$ contents in both liquid and CTA phases increase. Excessive addition of $TiO_2$ will lead to the large precipitation of CTA at this temperature.

(3) A small $TiO_2$ addition (e.g., 5%) may be considered for the refining of Ti-bearing steel grades to improve the melting properties of slag and the yield of Ti alloys, while further studies are needed in the near future.

**Author Contributions:** Conceptualization, X.Z. and Z.D.; methodology, X.Z. and Z.Y.; formal analysis, X.Z.; investigation, X.Z. and Z.Y.; resources, Z.D. and M.Z.; writing—original draft preparation, X.Z.; writing—review and editing, Z.D.; supervision, Z.D.; project administration, M.Z. and Z.D.; funding acquisition, M.Z. and Z.D. All authors have read and agreed to the published version of the manuscript.

**Funding:** This research was funded by The National Natural Science Foundation of China under Grant Nos. U20A20272 and 52074073.

**Data Availability Statement:** No data available.

**Conflicts of Interest:** The authors declare no conflict of interest.

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
