# Peer review of "Effect of TiO2 Addition on the Melting Behaviors of CaO-SiO2-30%Al2O3-5%MgO System Refining Slags"

_metals, doi:10.3390/met13020431_

Round 1

Reviewer 1 Report

Title: Effect of TiO2 addition on the melting behaviors of CaO-SiO2-30%Al2O3

 5% MgO system refining slags

·       What is the need to add TiO2?  Please write in the abstract.  

·       ‘In the case of Ti-bearing steel grades’:  Please write the amount of Ti in steel. 

·       Please explain the highest difference in the melting temperature at slag basicity of 2 (Figure 3).  Please elaborate on the higher melting temperature of melted slag than the mixed slag when TiO2 content is 10% with basicity 6 (Figures 4, 8, 9, 11 and 12).       

·       Please go through the conclusions after revision.           

Reviewer 2 Report

1. It is important to discuss the melting behavior of your sample as a function of your parameter by plotting and comparing data from literatures to improve your paper quality and reasonability. This work is very significant for understanding the melting behavior of TiO2-containing system as the effect of TiO2 content on melting point change is not remarkable (Figure 4), indicating that reproducibility should be considered again, although the author has measured melting point three times.

2. Clear discussion on the transition of melting behavior at specific basicity (C/S=6) and TiO2 content (5%) is the key, but it’s a bit unclear to explain it in the discussion section. Considering the network structure would be helpful as you already mentioned in the literature review. Furthermore, the author has tried to explain the combination of the melting temperature of the phases from reported literatures and qualitative fraction of solid phases. Here, it should be clear that what which factor mainly contribute to melting behavior between them. The "quantitative analysis" of solid fraction has to be considered if the author has though that solid fraction of precipitated phases is important.

3. Explain the reason why 1310 oC temperature for phase precipitation you have chosen.

4. The author has to show your sample’s shape change with experimental parameter.

Round 2

Reviewer 2 Report

The revised paper is well considered by following reviewer's comments and is allowed to be published.